# Development of Low-Cost Single-Chip Automotive 4D Millimeter-Wave Radar

**DOI:** 10.3390/s25154640

**Published:** 2025-07-26

**Authors:** Yongjun Cai, Jie Bai, Hui-Liang Shen, Libo Huang, Bing Rao, Haiyang Wang

**Affiliations:** 1Hangzhou City University Binjiang Innovation Center, Hangzhou 310051, China; 2College of Information Science & Electronic Engineering, Zhejiang University, Hangzhou 310027, China; 3School of Information and Electrical Engineering, Hangzhou City University, Hangzhou 310015, China; 4Suzhou Millimeter-Wave Technology Co., Ltd., Suzhou 215300, China; 5Chery New Energy Automobile Co., Ltd., Wuhu 241000, China

**Keywords:** 4D millimeter-wave radar, MIMO antenna array, azimuth, elevation

## Abstract

Traditional 3D millimeter-wave radars lack target height information, leading to identification failures in complex scenarios. Upgrading to 4D millimeter-wave radars enables four-dimensional information perception, enhancing obstacle detection and improving the safety of autonomous driving. Given the high cost-sensitivity of in-vehicle radar systems, single-chip 4D millimeter-wave radar solutions, despite technical challenges in imaging, are of great research value. This study focuses on developing single-chip 4D automotive millimeter-wave radar, covering system architecture design, antenna optimization, signal processing algorithm creation, and performance validation. The maximum measurement error is approximately ±0.2° for azimuth angles within the range of ±30° and around ±0.4° for elevation angles within the range of ±13°. Extensive road testing has demonstrated that the designed radar is capable of reliably measuring dynamic targets such as vehicles, pedestrians, and bicycles, while also accurately detecting static infrastructure like overpasses and traffic signs.

## 1. Introduction

With the rapid development of autonomous driving technology, on-board sensors, as the core components for environmental perception in vehicles, directly determine the safety and intelligent performance of autonomous driving systems. As the technology evolves toward higher-level autonomy, enhancing environmental perception capabilities has become a critical pathway to overcoming safety bottlenecks in complex scenarios. Among various sensors, millimeter-wave radar stands out as a pivotal technological solution in autonomous driving due to its all-weather operational capability, high-precision detection, and cost-effectiveness [1].

Compared to other sensors, millimeter-wave radar exhibits significant advantages:Cost Advantage: Millimeter-wave radar is more cost-effective than LiDAR, making it better suited for mass deployment in automotive and industrial applications.Environmental Adaptability: Unlike cameras, millimeter-wave radar operates independently of ambient light and maintains stable performance in complex weather conditions (e.g., fog, rain, snow).High Reliability: With robust anti-interference capability, millimeter-wave radar ensures consistent functionality even in electromagnetically noisy environments, such as urban areas with dense wireless signals.

Traditional 3D millimeter-wave radars (capable of measuring distance, azimuth angle, and velocity) often fail to distinguish objects due to the absence of target height information [2]. For example, they struggle to differentiate between manhole covers on the ground and road signs in the air, or between stationary soda cans and elevated bridge piers, leading to frequent occurrences of typical accidents (such as the 2016 Tesla-truck collision incident). Upgrading from traditional 3D to 4D millimeter-wave radars introduces the capability to measure target height (pitch angle), thereby enabling four-dimensional information perception of targets: distance, azimuth angle, velocity, and height. The ability to provide target height information is critical for intelligent driving vehicles to determine drivable areas. For instance, 4D millimeter-wave radars can detect overpasses, road signs, and ground obstacles (such as soda cans and manhole covers) ahead, helping vehicles avoid potential hazards. This significantly enhances the spatial contour parsing capability of obstacles, making it a key direction for upgrading intelligent driving sensor architectures [3,4].

In-vehicle radar systems are highly cost-sensitive, making single-chip solutions a research focus due to their superior cost–performance ratio. While traditional multi-chip cascaded solutions offer robust performance, their higher costs hinder large-scale adoption. In contrast, the direct upgrade of single-chip 3D radar to 4D radar presents technical challenges in achieving optimal imaging performance, yet it enables the acquisition of elevation data for detected objects. This capability allows advanced prediction of potential hazards like overpasses and traffic signs along the road. Through technological innovation and cost optimization, single-chip 4D millimeter-wave radar is poised to play an increasingly vital role in intelligent driving systems [5,6,7].

The main contributions of this work can be summarized as follows:(1)Single-Chip Design: We propose a design scheme for a single-chip 4D millimeter-wave radar, realizing a low-cost and high-performance 4D radar system through a highly integrated hardware platform and an optimized sparse MIMO antenna array. We developed high-gain, low-sidelobe antenna arrays to enhance detection capability and anti-interference performance. This design breaks through the cost constraints of traditional multi-chip cascade solutions, offering the potential for large-scale commercial applications.(2)Rain Clutter Identification: A rain clutter identification method based on the distribution characteristics of raindrops is proposed. By statistically analyzing the speed and distance distribution of raindrops, this method effectively distinguishes between raindrops and real targets, thereby reducing the interference of rain clutter on radar detection.(3)Noise Point Suppression: A noise-suppression method based on angular FFT peak-amplitude variance is proposed. By designing an effective strategy, this approach suppresses noise point interference while preserving true targets, thereby enhancing radar target-detection stability in complex environments.

The structure of this paper is as follows: Section 2 presents the system hardware design, including antenna radiation patterns. Section 3 details the signal processing flow with angle accuracy analysis, along with rain-clutter suppression and noise point suppression methods. Section 4 provides extensive field test results validating system effectiveness. Section 5 discusses implementation approaches for 4D radar and future development trends. Section 6 concludes the paper.

## 2. Radar System Hardware Platform and Beam Design

The radar is connected to the vehicle body gateway via the public CAN bus to acquire information such as vehicle speed, gear position, turn signals, door status, yaw rate, and steering wheel angle. Meanwhile, it uploads target information, dangerous target warning information, and diagnostic fault information to the vehicle body CAN network. The overall dimensions of the hardware structure are 109 mm × 67 mm × 34.7 mm (including the connector), with a weight of approximately 86 g.

The hardware circuit section mainly consists of the following components:(1)RF and Signal Processing Circuit

Functions: Processes radio frequency signals (e.g., millimeter-wave radar signals), performs analog-to-digital conversion (ADC), and implements baseband signal processing algorithms (e.g., FFT, CFAR, and target detection).

(2)Power Management Circuit

Functions: Converts vehicle battery power (12 V/24 V) to stable voltages (e.g., 3.3 V, 1.8 V) for RF chips, microcontrollers (MCUs), and sensors. Includes surge protection, EMI filtering, and low-power sleep modes.

(3)CAN Transceiver Circuit

Functions: Bridges the digital logic of the radar MCU to the vehicle’s CAN bus. Converts differential CAN signals to TTL/CMOS levels for data transmission (e.g., vehicle speed, danger warnings) and reception (e.g., steering angle).

(4)Peripheral Storage Circuit

Functions: Stores configuration parameters (e.g., radar calibration data), fault codes (DTCs), and temporary target logs. Typically uses EEPROM (for non-volatile storage) or SPI Flash (for large data buffers).

Specifically, for the RF circuit, the system employs the AWR2944 chip, which is built with TI’s low-power 45 nm RFCMOS process and enables unprecedented levels of integration in a small form factor and minimal BOM. The chip integrates essential RF components including a PLL, VCO, mixer, and ADC, supporting multi-channel antenna interfaces for advanced radar operations. Operating in the 76–81 GHz frequency band with over 4 GHz contiguous bandwidth, it features a 4-transmit (TX) and 4-receive (RX) channel architecture, facilitating high-resolution MIMO (Multiple-Input Multiple-Output) signal acquisition. For signal processing, the system leverages a 360 MHz C66x digital signal processor combined with the HWA 2.1 radar hardware accelerator, which optimizes real-time execution of key algorithms such as FFT (Fast Fourier Transform), CFAR (Constant False Alarm Rate), log magnitude, and memory compression for accurate target detection and tracking in automotive environments. This highly integrated design ensures compact form factor compatibility and robust performance for advanced driver assistance systems (ADAS).

The radar antenna employs a MIMO architecture to minimize system volume, featuring a comb-shaped radiator for compact integration. To enhance the comb antenna’s gain, we optimized the geometry of individual radiating elements:

Structural Optimization: Increasing the effective radiation area through specialized shape and size optimization (simulated via HFSS), simultaneously improving the antenna’s radiation efficiency.

Current Distribution Control: Current distribution weighting optimization suppressed sidelobe levels to −15 dB (vs. −12 dB for unoptimized arrays), achieved through adaptive current amplitude/phase weighting.

As shown in Figure 1, the optimized antenna achieves a 60° 3 dB beamwidth in the horizontal direction for wide angular coverage and simultaneous multi-target detection, along with a 16° 3 dB beamwidth and 20° 6 dB beamwidth in the elevation direction, ensuring accurate height estimation for overpassing vehicles.

## 3. Signal Processing

### 3.1. FMCW Radar Signal Processing and Angle Accuracy

The FMCW is a popularly used waveform for automotive radar and other applications. The fast chirp ramp sequence is shown in Figure 2 [8]. The process of obtaining range and velocity utilizes 2-DFT which is performed first for each individual chirp/ramp to obtain range information and second across ramps to obtain velocity information. By further computing angle information based on the antenna configuration used to receive the chirp ramp sequence, the 3D target data can be constructed and consists of range, velocity, and angle [9].

Frequency-Modulated Continuous Wave (FMCW) waveform [10], also known as a chirp, is a complex sinusoidal signal whose frequency increases linearly with time *t*, expressed as [11]:(1)$$\begin{array}{c} \;{\;f}_{T}(t)=f_{c}+\frac{B}{T}t \end{array}$$
where B is the bandwidth, T is the chirp period, and $f_{c}$ is the carrier frequency. The received signal is mixed with the transmitted chirp to generate a complex sinusoidal beat frequency signal, whose frequency equals $f_{b}=f_{R}+f_{D}$, where $f_{R}$ is the range frequency, $f_{D}=(2v/c)f_{c}$ is the doppler frequency, *c* is speed of light. The beat frequency signal is passed through a low-pass filter (LPF) in the RF domain to remove out-of-band interference, then digitized. By applying a Fast Fourier Transform (FFT) to the beat frequency signal sampled along fast time (within a single chirp), $f_{R}$ can be extracted. Based on $\;f_{R}$, the target distance *R* is calculated using(2)$$\begin{array}{c} \;R=\frac{c\cdot f_{R}\cdot T}{2B} \end{array}$$

To obtain the doppler velocity of a target, a second Fast Fourier Transform (FFT) is performed along the slow-time dimension (across consecutive chirps), where the range frequency $f_{R}$ remains consistent within the same range bin [12].(3)$$\begin{array}{c} \;v=\frac{\lambda }{4\pi }\cdot \frac{∆\phi }{T} \end{array}$$
where λ is the operating wavelength, ∆ϕ denotes the phase difference between two adjacent chirps.

The antennas transmit waveforms in a manner that ensures their orthogonality. At each receive antenna, for a system with Mt uniformly spaced transmit antennas and Mr uniformly spaced receive antennas, a virtual array with MtMr elements can be synthesized. The array response can be expressed as(4)$$\begin{array}{c} \;y=A(\theta )s+n \end{array}$$
where $A(\theta )=[\alpha (\theta_{1}),\ldots ,\alpha (\theta_{k})]$ is the control matrix of the virtual array $\alpha (\theta_{k})=[1,e^{j\frac{2\pi }{\lambda }d\sin(\theta_{k})},\ldots ,e^{j\frac{2\pi }{\lambda }(M_{t}M_{r}-1)d\sin(\theta_{k})}{]}^{T}$, *d* is the minimum spacing of the synthetic virtual array, $\theta_{k}$ denotes the angle of the *k*-th target, n represents the noise term, and s denotes the echo signal of the *k*-th target, angle estimation can be performed by applying a Fast Fourier Transform (FFT) to the snapshots of the array elements. When employing non-uniform or sparse linear arrays combined with MIMO radar technology, the key challenges lie in the selection of array element positions and the utilization of virtual sparse arrays for angle measurement [13].

Let the azimuth angle and elevation angle be denoted as *θ* and *φ*, respectively, which satisfy the following relationship:(5)$$\begin{array}{c} \;sin(\alpha )=sin(\theta )cos(\varphi ) \end{array}$$
where α  is defined as the coupled azimuth-elevation composite angle. Similar to the doppler processing, the frequency variation between different receiving antennas can be ignored. This simplification allows for the calculation of the target’s angle based on the phase difference between the antennas.

The phase difference between adjacent elements can be expressed as follows [14]:(6)$$\begin{array}{c} \;∆\varphi =\frac{2\pi dsin(\theta )cos(\varphi )}{\lambda } \end{array}$$

After performing a Fast Fourier Transform (FFT) along the antenna dimension, the angle is further derived from the peak of the spectrum corresponding to ∆φ.

The synthetic virtual array of this system is shown in Figure 3. It comprises four distinct colors, with each color representing a virtual antenna synthesized from different transmitting antennas. Both the horizontal and vertical minimum spacings are *d*, equivalent to half-wavelength. The following sections detail the following components:
Formation of the horizontal and elevation arrays;Derivation methodology for target azimuth and elevation angles.

For each target passing CFAR detection, data from its 16 virtual channels are first extracted. The row containing the highest number of horizontal channels is selected as the horizontal array for azimuth measurement. The included channels are [T2R4, T3R4, T2R2, T3R2, T2R1, T4R4, T3R1, T4R2, T4R1]. Zeros are then padded at vacant grid positions to form the final sparse horizontal array: [T2R4, 0, T3R4, 0, 0, T2R2, 0, T3R2, 0, 0, 0, 0, 0, 0, 0, 0, T2R1, T4R4, T3R1, 0, 0, 0, T4R2, 0, 0, 0, 0, 0, 0, 0, 0, 0, 0, T4R1]. The total aperture length is 33*d*. When the target is at 0° azimuth, the beam pattern exhibits a 3.1° 3 dB beamwidth with a sidelobe level of approximately −5 dB.

Following FFT operations, peak detection, and parabolic interpolation on the sparse horizontal array, the target’s coupled azimuth-elevation composite angle α is obtained, requiring decoupling to resolve individual angles. We therefore prioritize elevation calculation before azimuth estimation. Per Figure 3, up to four channels are selected for the elevation array: first, the column with maximum channels [T1R2, T3R4, T2R3] is chosen, then T1R3 is added to form [T1R2, T1R3, T3R4, T2R3]. After zero-padding vacant grid positions to create the sparse elevation array [T1R1, 0, 0, T1R3, 0, 0, 0, 0, 0, T3R4, 0, 0, T2R3], phase compensation is essential for T1R3 due to its 3-grid horizontal offset to finalize the elevation array for angle measurement, with the required compensation values as follows:(7)$$\begin{array}{c} \;\phi =3\cdot \frac{2\pi dsin(\alpha )}{\lambda } \end{array}$$

When the target is positioned at 0° elevation, the beam pattern exhibits a 3 dB beamwidth of 6.1° with corresponding sidelobe levels approximately at −5 dB. Subsequently, FFT operations, peak detection, and parabolic interpolation are performed on the elevation array to compute the elevation angle φ. This elevation angle is then substituted alongside the coupled composite angle α into Formula (5) to derive the azimuth angle, thereby resolving both angular dimensions.

In the chamber, a corner reflector is positioned directly in front of the radar, with the radar elevation maintained at 0°. The actual azimuth of the corner reflector is compared to the measured azimuth by mechanically rotating the radar in increments of 2°. Similarly, the elevation of the corner reflector is measured when the azimuth is set to 0°. The error curve illustrating the difference between the measured values and the actual azimuth and elevation of the corner reflector is presented in Figure 4.

The azimuth error is less than 0.3° within the angular range of −70° to 70° and no more than 0.2° within the range of ±30°. Additionally, the elevation error is less than 0.4° within the range of ±13°. Therefore, the antenna array designed in this scheme demonstrates high measurement accuracy in angle measurement.

### 3.2. Clutter Suppression

#### 3.2.1. Mitigation and Avoidance of Rain Clutter Interference

While millimeter-wave radar can penetrate rain and fog to detect distant targets, when rainfall intensifies to a detectable level by the radar, raindrops are often mistakenly identified as targets, triggering false alarms. These alarms are generally undesirable, as continuous rainfall scenarios may cause persistent radar alerts suggesting approaching targets—even when the vehicle is in an open area with no actual obstacles.

Current public literature indicates there are no highly effective methods for mitigating rain clutter interference, although individual radar or automotive manufacturers may employ proprietary solutions [15,16,17]. This paper proposes a raindrop identification method based on raindrop distribution characteristics to alleviate rain clutter interference.

As raindrops fall from several kilometers in altitude, they are subject to the combined effects of gravity, air resistance, and buoyancy. The terminal velocity calculation formula is as shown [18]:(8)$$\begin{array}{c} \;v_{t}=\sqrt{\frac{2mg}{\rho AC_{d}}} \end{array}$$
where m represents the mass of the raindrop, g denotes gravitational acceleration, ρ is air density, A  is the raindrop’s frontal area, and $C_{d}$ is the drag coefficient (approximately 0.47 for spherical raindrops).

The relationship between raindrop mass and radius can be expressed as(9)$$\begin{array}{c} \;m=\frac{4}{3}\pi r^{3}\rho_{\omega } \end{array}$$
where $\rho_{\omega }$ represents the density of the raindrop, and the frontal area of the raindrop is given by(10)$$\begin{array}{c} \;A=\pi r^{2} \end{array}$$

Therefore, the terminal velocity of the raindrop can be derived as(11)$$\begin{array}{c} \;v_{t}=\sqrt{\frac{2\cdot \frac{4}{3}r^{3}\cdot \rho_{\omega }\cdot g}{\rho \cdot \pi r^{2}\cdot C_{d}}}=\sqrt{\frac{8r\rho_{\omega }g}{3\rho C_{d}}} \end{array}$$

In typical rainfall conditions, raindrop radii are generally less than 4 mm (mostly below 2 mm). Consequently, under low wind conditions, the terminal velocity of raindrops rarely exceeds 10 m/s. Since radar measurements are conducted in near-horizontal directions (capturing the horizontal component of falling raindrops), this component is primarily influenced by wind speed and terrain [18,19]. Under normal conditions with mild winds, we assume raindrops have a maximum inclination angle of approximately 30°. Under this constraint, the maximum horizontal component of raindrops at terminal velocity reaches $v_{level}=v\cdot \tan(25{}^{\circ})\approx 4.66\;m/s$. We therefore set ±4 m/s as the threshold for radar target detection.

Given that rainfall distribution is spatially uniform within limited areas (e.g., 100 m × 100 m) at any given moment, raindrop sizes remain consistent, and their terminal velocities are nearly identical. Furthermore, we restrict the radar detection range to within 30 m because distant low-speed targets pose no collision risk and thus require no alerts—only nearby moving objects are of concern. By statistically analyzing raindrops within this defined region (2–30 m), we can derive their velocity distribution and variance, enabling effective clutter discrimination.

We collected rainfall data under normal rainy conditions using a compact vehicle equipped with a commercial millimeter-wave radar, which was stationed alongside an open road during precipitation. The data acquisition scenario is illustrated in Figure 5.

During data collection, the radar was mounted at a height of approximately 60 cm. Data acquisition was performed while the radar remained stationary. Throughout continuous rainfall conditions, we collected approximately 40 min of data comprising 35,159 frames. The velocity distribution and distance distribution of the detected raindrops is shown in Figure 6 and Figure 7. The majority of raindrops detected by the radar fall within the velocity range of −1 m/s to 1 m/s.

Meanwhile, the distance distribution of the detected raindrops is more uniform. Within our selected range of 2–30 m, the number of raindrop detections remains relatively consistent between 5 and 30 m, with an especially uniform distribution observed between 8 and 23 m. Therefore, we quantify raindrop characteristics using both velocity variance and distance variance. The variance is calculated as follows:(12)$$\begin{array}{c} \;\sigma^{2}=\frac{1}{N}\sum_{i=1}^{N}(x_{i}-\mu {)}^{2} \end{array}$$
where $\sigma^{2}$ represents the population variance, N is the sample size, μ is the mean value, and $x_{i}$ denotes either velocity or distance. Since the radar cannot consistently detect dynamic raindrops (i.e., the number of detected targets fluctuates dynamically under the same conditions due to the unstable reflection of radar waves), and rainfall typically persists for extended periods, we employ a sliding-window statistical method to process the data and avoid abrupt changes in algorithm output. Specifically, we create an array with data from a fixed number of frames (60 frames in this case). As new radar data are acquired, the array is updated by replacing the oldest data, enabling stable processing of time-series data and minimizing fluctuations caused by radar measurement variability.

During a rainfall test in Guangzhou, China, we obtained the results shown in Figure 8.

Over approximately 10 min of measurement, the algorithm output closely matched the trend of raindrop counts detected by the radar, confirming that the algorithm’s results align with precipitation intensity. This demonstrates the algorithm’s effectiveness in rainy conditions. For comparison, we also tested the algorithm in a more complex sunny scenario at a roadside in a small town, where pedestrians, bicycles, and cars frequently passed by. The results are shown in Figure 9.

Compared to the rainy scenario, the radar detected some point clouds with raindrop-like characteristics even in sunny conditions. However, the algorithm output remained at a low level (significantly lower than during rainfall), showing clear distinction between rainy and sunny conditions. This confirms the algorithm’s ability to avoid false classification in non-rainy scenarios.

#### 3.2.2. Noise Point Elimination Method in Complex Environments

In automotive radar point cloud measurements, beyond legitimate targets, there often exist numerous “extraneous” points that interfere with proper target tracking. In multi-lane traffic scenarios, mutual interference between vehicle radars or cross-interference from nearby systems can degrade performance and introduce data inaccuracies. While existing literature proposes various interference mitigation strategies, no comprehensive solution effectively eliminates all noise sources under extreme conditions [20,21,22,23]. This paper presents a simple yet efficient noise point elimination method that preserves genuine targets while suppressing most interference.

Legitimate targets typically exhibit high signal-to-noise ratio (SNR) characteristics. After applying angular-dimension FFT processing, their peaks significantly dominate adjacent positions, as illustrated in Figure 10. In contrast, noise points (e.g., ghost reflections from open ground surfaces, raindrops) demonstrate distinct angular FFT signatures; shown in Figure 11, their response curves remain relatively flat, with minimal amplitude differences between peak values.

Despite their lower amplitudes, these noise peaks may still pass through Constant False Alarm Rate (CFAR) detection thresholds and be misclassified as valid targets [24]. To address this, we propose evaluating the differential amplitudes between the top N peaks in angular FFT profiles. By setting appropriate thresholds for peak amplitude variance, our method effectively discriminates and filters out noise points while retaining true targets.

For the cases shown in Figure 10 and Figure 11, we calculate the amplitude differences between the first maximum peak and subsequent peaks (2nd, 3rd, and 4th), as detailed in Table 1:

Where Δp_1-2_ represents the amplitude difference between the 1st and 2nd peaks, Δp_1-3_ represents the amplitude difference between the 1st and 3rd peaks; Δp_1-4_ represents the amplitude difference between the 1st and 4th peaks.

Here, Δp_1-2_ denotes the amplitude difference between the 1st and 2nd peaks, Δp_1-3_ represents the 1st-to-3rd peak difference, and Δp_1-4_ indicates the 1st-to-4th peak difference. Analysis reveals that in the echo curves of human targets (legitimate targets), these three differential values are significantly greater than those observed in noise point echo curves. This characteristic enables effective noise point rejection through threshold-based filtering.

Building upon this methodology, we conducted road target detection tests using a millimeter-wave radar system, with the following parameters:Targets of interest: Electric two-wheelers and automobiles;Undesired interference sources: Ground clutter, multipath reflections, and vegetation-induced noise points.

The evaluation focused on the following:Identification rate of unwanted noise points;Misidentification probability of valid targets.

To validate the algorithm’s practical efficacy, the test scenario is illustrated in the figure below:

In Figure 12, red points indicate approaching objects (potential hazards); blue points represent receding objects; gray points denote stationary objects. Key areas are marked as follows:Zone 1: Valid two-wheeler targets;Zone 2: Multipath reflections from roadside vegetation edges;Zone 3: Ground clutter-induced noise points.

The system successfully identified over 90% of unwanted points in this scenario. Under consistent environmental conditions, we statistically evaluated the following: (1) correct identification rate for noise points; (2) false identification probability for legitimate targets. The quantitative results are presented in Table 2 and Table 3:

Measurement Note: The statistical data were collected under the following dynamic conditions: Pedestrians: Tracked while walking toward the radar from 30 m distance; Vehicles: Approaching the radar at 50 km/h from 70 m range; Two-wheelers: Moving toward the radar at 20 km/h from 40 m.

The statistical results demonstrate that our algorithm achieves the following:>70% ground clutter rejection rate;~90% vehicle multipath detection accuracy;<5% false identification rate for legitimate targets.

This confirms the algorithm’s exceptional effectiveness in noise suppression for automotive radar applications, successfully addressing three critical challenges: ground reflection interference, multipath artifacts, and target discrimination accuracy.

## 4. Radar Performance Test

The radar system has been integrated within the front bumper assembly of a compact SUV, as depicted in Figure 13. Real-time measurement data are transmitted to a portable computing unit for visualization, with output parameters encompassing Cartesian coordinates (X-Y), slant range, radial velocity, azimuth angle, and elevation angle relative to the sensor array. Extensive field validation was conducted across multiple road surfaces (pavement, gravel, dirt) to assess system performance under varying environmental conditions.

To validate the azimuth accuracy in the horizontal plane, an experimental setup was implemented involving two vehicles traveling in parallel trajectories. The test vehicles maintained a higher relative velocity compared to the radar-equipped platform and gradually diverged from the target vehicle during the trial.

The measurement results are illustrated in Figure 14.

Under a slant range of 5 m, the lateral separation between the two target vehicles was maintained at approximately 5 m. The measured Cartesian coordinates (*X*-axis) of the targets were recorded as follows:

Left vehicle: X_1_ = 4.4 m;

Right vehicle: X_2_ = 0.6 m.

Both vehicles traveled in straight trajectories within their respective lanes. When the slant range increased to 100 m, the actual lateral separation remained consistent at 5 m. Corresponding coordinate measurements at this extended range showed the following:

Left vehicle: X_1_ = 4.4 m;

Right vehicle: X_2_ = 0.8 m.

The right-side target exhibited a lateral displacement of approximately 0.2 m, corresponding to an angular deviation of 0.1°, which aligns with the angular error margin quantified in Section 3.1. Notably, at a slant range of 100 m, the radar system successfully resolved both targets with an angular separation of approximately 3.0°, demonstrating robust angular resolution performance under extended-range conditions.

During road testing, the radar achieved a maximum detection range of 210 m, effectively identifying and tracking large trucks, cars, electric bicycles, and pedestrians, whether vehicles were operating at low or high speeds. Additionally, the radar clearly detected guardrails and roadside greenery belts on both sides of the road. As shown in Figure 15, even when vehicles were traveling at high speeds, the radar successfully detected them.

The radar sensor is established as the spatial origin of the coordinate system. As depicted in Figure 15, seven vehicles are positioned in front of the sensor array, with the farthest target located at a maximum slant range of 210 m. Within the 50 m slant range zone, three vehicles are clustered in adjacent lanes. The roadside metal fences manifest as sparsely distributed point clouds, exhibiting vertical alignment despite lower density compared to 4D imaging radars. By integrating spatial coordinates (X-Y-Z) and radial velocity data, static target classification is achievable through height reconstruction and aspect-angle analysis.

As illustrated in Figure 16, the radar system accurately renders overpass structures and metal signage within its field of view, demonstrating robust clutter suppression and high-resolution imaging capability.

When encountering suspended targets such as overpasses or traffic lights above the road, these structures manifest as detectable point clouds directly ahead of the vehicle. This enables high-altitude target identification through height, position, and velocity parameters extracted from the point cloud. As demonstrated in Figure 16, when approaching a road section with an overhead footbridge, the bridge’s reflection point cloud (highlighted in the yellow box) is clearly visible. The point cloud’s horizontal orientation corresponds to the footbridge’s actual direction, as shown in the image. Simultaneously, vehicles traveling adjacent to the footbridge are clearly resolved. The footbridge’s structural height ranges between 5 and 10 m. With the radar fixed at the vehicle’s front, the detected bridge height exhibits minimal variation. As the vehicle approaches, the target’s reflected point cloud expands proportionally, with both the bridge and adjacent vehicles clearly captured in Figure 16.

As previously mentioned, the unique array design makes the system’s elevation measurement capability a key highlight. With an elevation angle resolution of 6.1°, this radar can fulfill conventional height measurement requirements where extreme precision is not critical. It serves as a cost-effective alternative to 8T8R (8-transmit/8-receive) or higher-channel 4D radars. Qualitative and intuitive road test results are presented in Figure 15 and Figure 16. To quantitatively demonstrate the radar’s height measurement capability, targets at various heights were tested to evaluate the radar’s distinguishability through elevation angle measurements.

We separately evaluated the elevation angle performance for overpasses and stationary vehicles. Raw points within the ego lane were selected for 2D visualization, comparing the variation in height and lateral distance with longitudinal distance. Results are shown in Figure 17, where target height performance is only displayed within 80 m. Beyond 80 m, the limited elevation angle resolution (6.1°) and multipath reflections from the ground cause the road surface to act as a guardrail-like reflector. This generates ghost targets below ground level—analogous to false targets outside guardrails—rendering the radar unable to fully resolve actual elevated targets from subsurface ghosts. Numerous targets near 0° elevation appear, making overpasses and vehicles indistinguishable. As shown, targets within 80 m are separable via elevation angle, providing essential information for drivable area detection.

We also evaluated the elevation performance for pedestrians and ground-mounted stationary corner reflectors, with results shown in Figure 18. Only scenarios within 20 m are displayed, as beyond this range the elevation angle of corner reflectors approaches 0°—insufficient to permit reliable discrimination. Notably, as distance decreases, the elevation angle magnitude of corner reflectors progressively increases and pedestrian elevation angles remain comparatively stable. Critically, corner reflectors exhibit consistently negative elevation values, while pedestrians predominantly show positive elevation readings. This enables robust differentiation between ground-level static targets and above-ground pedestrians through elevation analysis.

## 5. Discussion

The current implementation methods for 4D millimeter-wave imaging radar primarily include the following:(1)Chip cascade technology, exemplified by the AWR2243 chip cascade scheme from Texas Instruments (TI), features a single chip equipped with three transmitters (TX) and four receivers (RX). By cascading four chips, a configuration of 12 TX and 16 RX is achieved, resulting in a total of 192 virtual channels.(2)A single-chip design enables the transmission and reception of multiple signals on a single chip. A notable example is Arbe’s chipset, which can be expanded to accommodate 48 transmitters (TX) and 48 receivers (RX), generating a total of 2304 virtual channels. In contrast, the radar-on-chip solution from the US company Uhnder utilizes a 12 TX/8 RX configuration, capable of forming 96 virtual channels.

However, all of the above-mentioned solutions face challenges such as high cost, complex technology, and long development cycles, and they have not been widely adopted in the market. Although it is relatively difficult to achieve the ideal imaging effect with the method of directly upgrading 3D radar to 4D radar, this method can obtain the height information of the target, enabling the early prediction of obstacles on the road such as overpasses and road signs. In most cases, it is meaningful for the drivable area detection of forward-looking radars. Moreover, the low cost facilitates large-scale application. Based on this concept, this paper proposes a design scheme for a single-chip 4D millimeter-wave forward-looking radar. It achieves height measurement through the design of sparse MIMO antennas. In road tests, the developed radar effectively detected high-position targets such as traffic lights and overpasses. Compared with the standard 3D millimeter-wave radar, this scheme has a cost advantage, lower development difficulty, and accelerates the product commercialization process. Therefore, we believe that the 4D single-chip radar will become the main direction of future forward-looking radar systems and will be widely installed in vehicles.

## 6. Conclusions

This paper details the development process of a single-chip four-dimensional (4D) automotive millimeter-wave radar, encompassing system architecture design, antenna optimization, signal processing algorithm creation, and performance verification. It primarily addresses the issue that traditional 3D millimeter-wave radars lack target height information. By upgrading to a 4D radar, it achieves four-dimensional information perception of target distance, azimuth, velocity, and height, significantly enhancing obstacle detection capabilities and safety in autonomous driving.

The radar system employs a highly integrated hardware platform. Through the optimized design of a sparse MIMO antenna array and efficient signal processing algorithms, it can accurately measure both dynamic and static targets in complex environments. In actual road tests, it demonstrates high detection accuracy and reliability.

In addition, this paper proposes an effective rain clutter recognition method based on the characteristics of raindrop distribution and a method for suppressing noise points in complex environments based on the amplitude differences of angle FFT peaks, further improving the robustness of the radar system.

After extensive road-test verification, the designed radar can reliably detect dynamic targets such as vehicles, pedestrians, and bicycles, as well as static infrastructure such as overpasses and traffic signs. It has significant advantages in terms of cost, performance, and commercialization, and can be reliably applied in the field of autonomous driving, with broad market prospects.

## Figures and Tables

**Figure 1 sensors-25-04640-f001:**
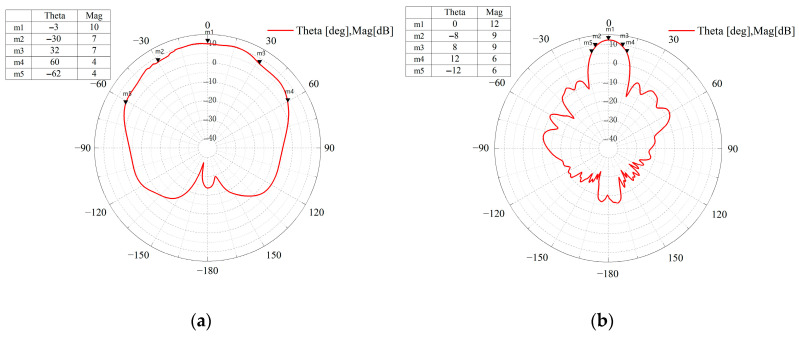
Radar system transmitting antenna beam pattern: (**a**) horizontal direction; (**b**) elevation direction.

**Figure 2 sensors-25-04640-f002:**
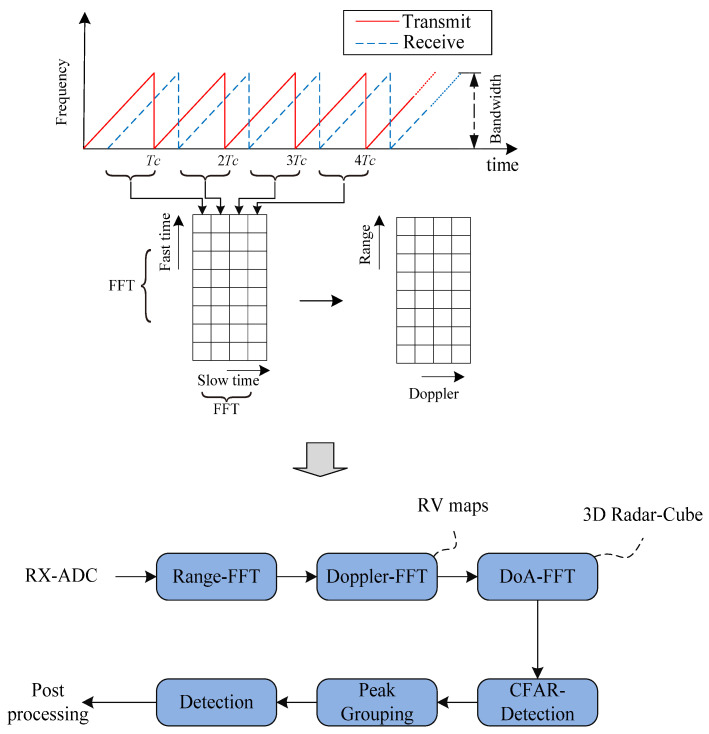
An example of the fast chirp ramp sequence waveform for range and velocity detection and a radar signal processing process.

**Figure 3 sensors-25-04640-f003:**
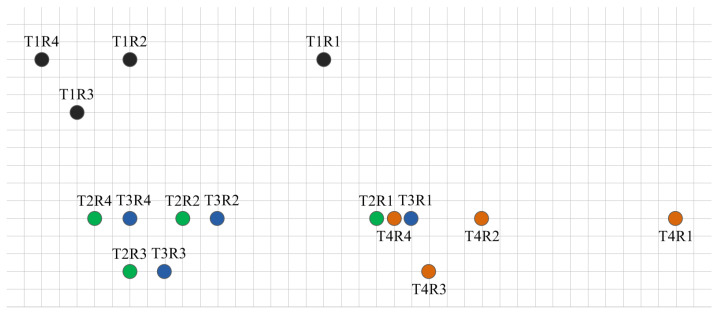
Synthetic virtual array of 4T4R. Black, green, blue, and brown represent the antennas virtualized from transmitting antennas T1, T2, T3, and T4 respectively.

**Figure 4 sensors-25-04640-f004:**
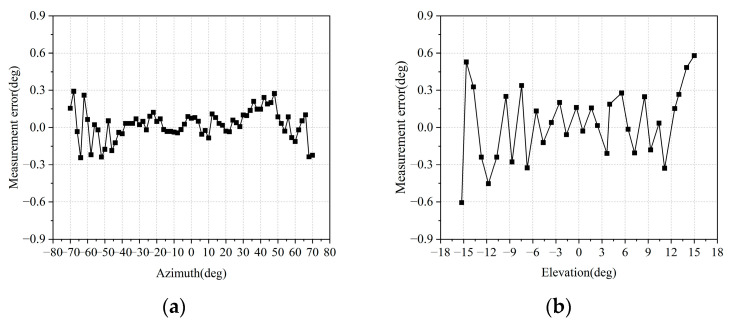
Angle error curve for measuring corner reflector: (**a**) azimuth angle; (**b**) elevation angle.

**Figure 5 sensors-25-04640-f005:**
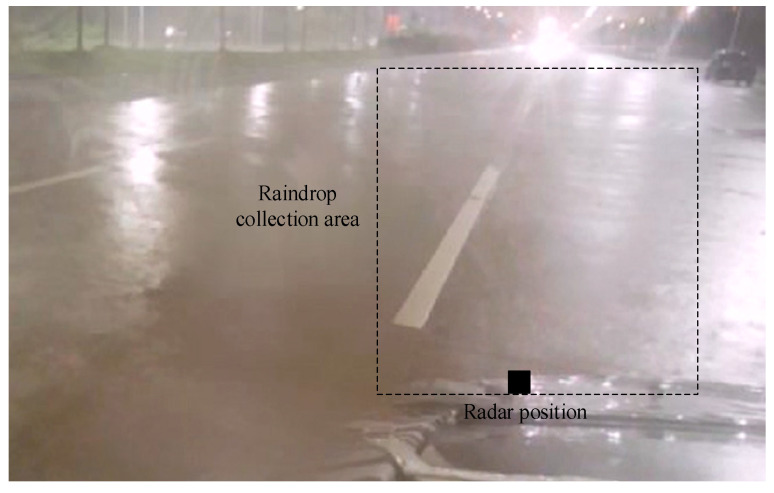
Raindrop measurement location and study region.

**Figure 6 sensors-25-04640-f006:**
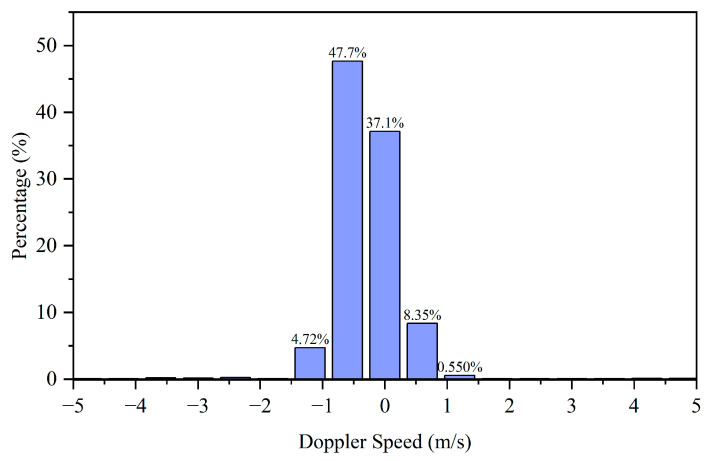
Velocity distribution of radar-detected raindrops.

**Figure 7 sensors-25-04640-f007:**
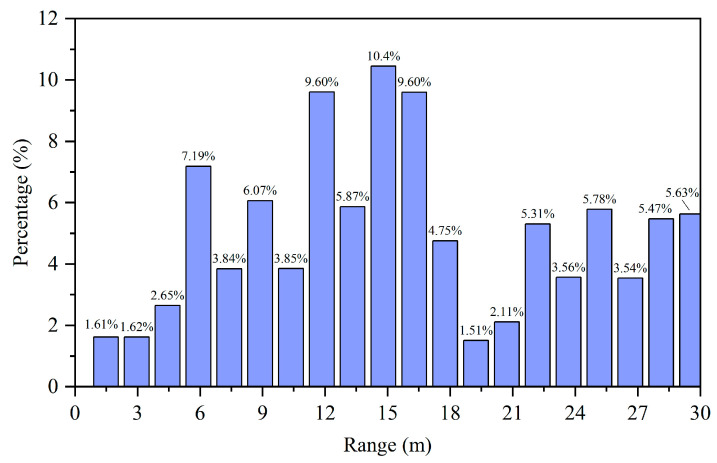
Distance distribution of radar-detected raindrops.

**Figure 8 sensors-25-04640-f008:**
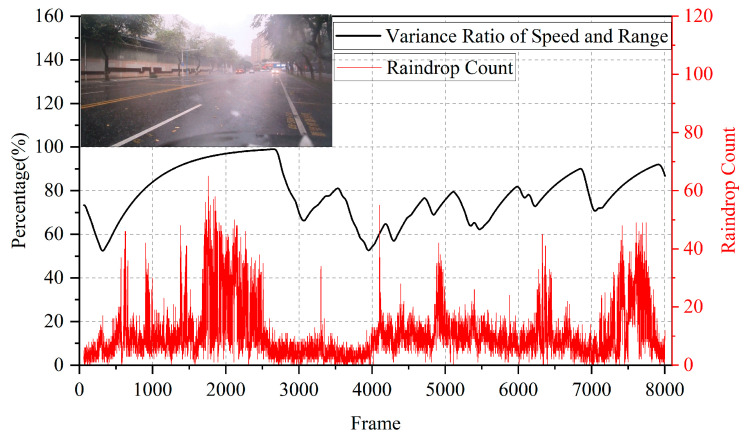
Rainfall scenario (gray curve: number of raindrop point clouds measured by radar; red curve: algorithm output results (percentage scale)).

**Figure 9 sensors-25-04640-f009:**
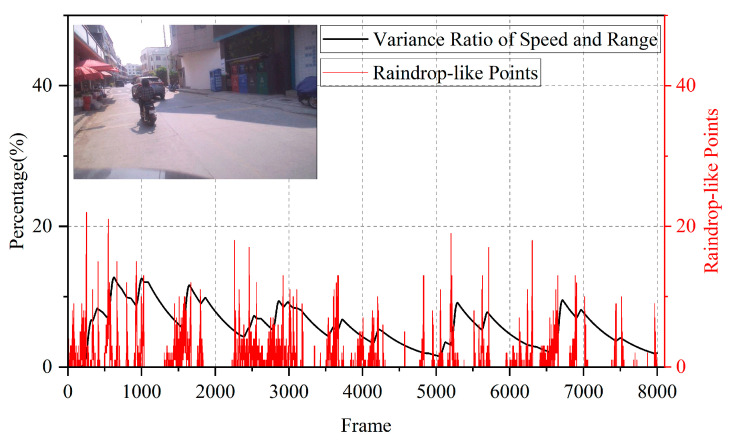
Sunny urban roadside scenario (gray curve: detected target point clouds measured by radar; red curve: algorithm output results (percentage scale)).

**Figure 10 sensors-25-04640-f010:**
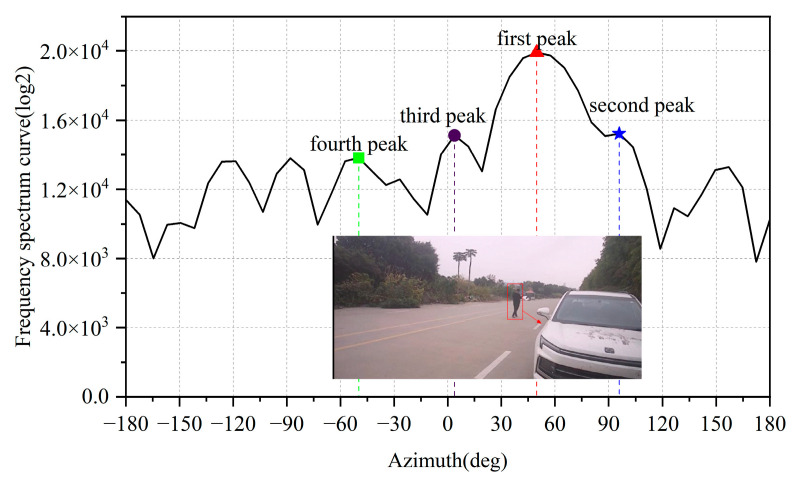
Angular FFT profile of normal targets.

**Figure 11 sensors-25-04640-f011:**
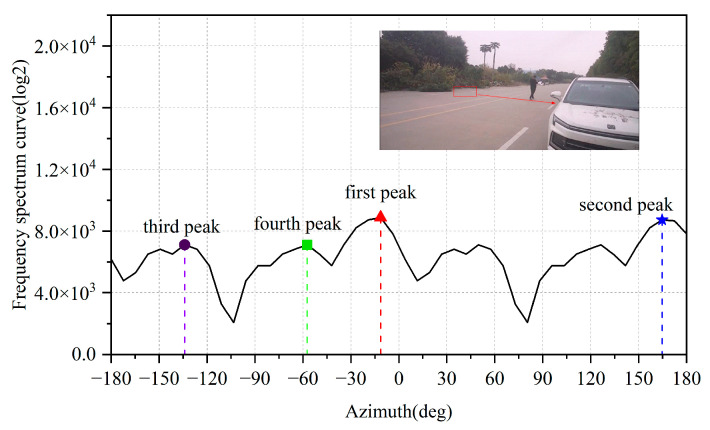
Angular FFT profile of clutter in open areas.

**Figure 12 sensors-25-04640-f012:**
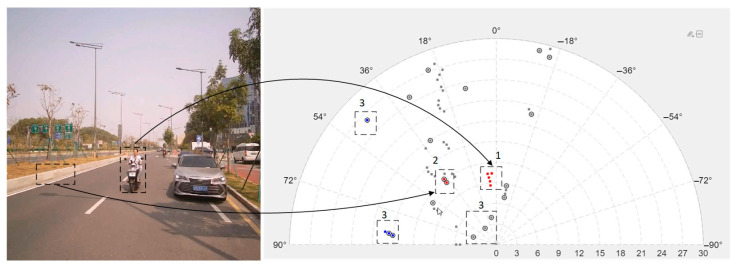
Test scenarios and point cloud distributions. Red dots, blue dots, and gray dots represent approaching targets, receding targets, and stationary targets respectively.

**Figure 13 sensors-25-04640-f013:**
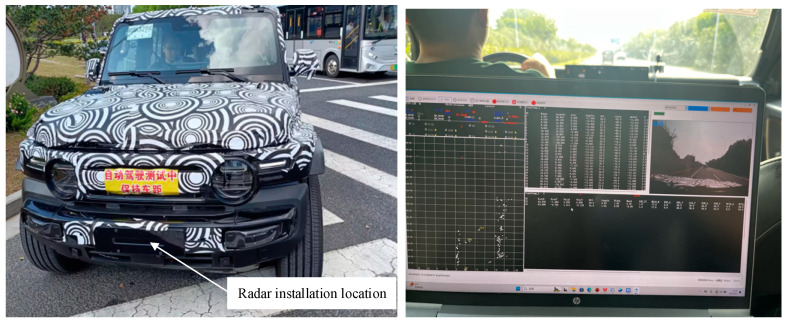
Radar installation location and computer interface.

**Figure 14 sensors-25-04640-f014:**
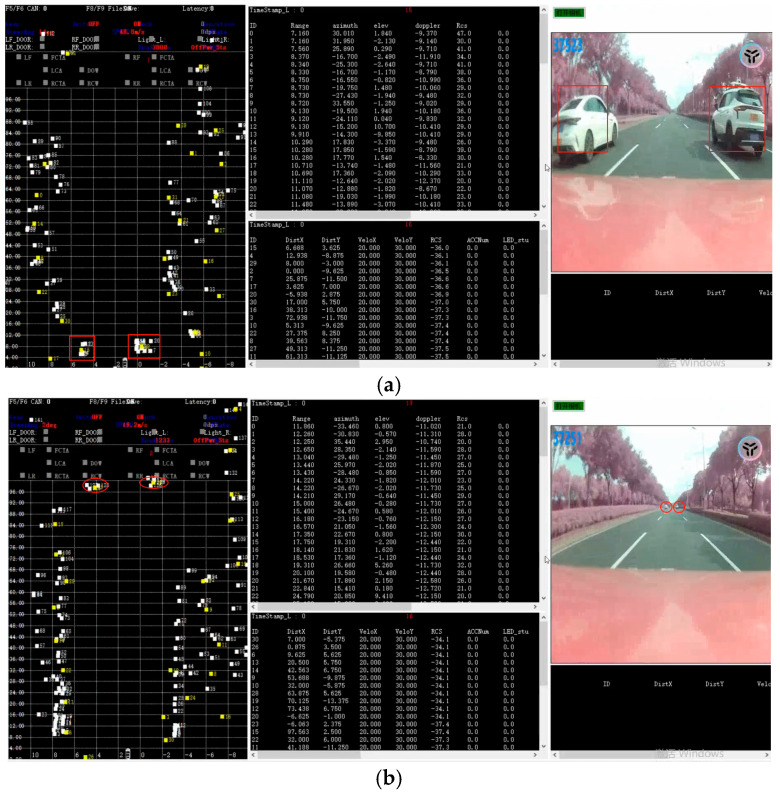
Test results for two targets in adjacent lanes at longitudinal distances of 5 m (**a**) and 100 m (**b**) ahead of the radar.

**Figure 15 sensors-25-04640-f015:**
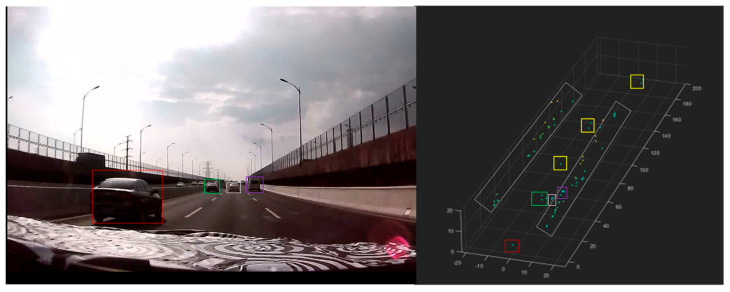
The display effect of vehicles during road testing, with nearby vehicles indicated by red, green, white, and purple boxes, and more distant vehicles marked with yellow boxes. The gray area represents the fence on both sides. The green dots indicate ground-level targets, while the yellow dots indicate aerial targets at high altitudes.

**Figure 16 sensors-25-04640-f016:**
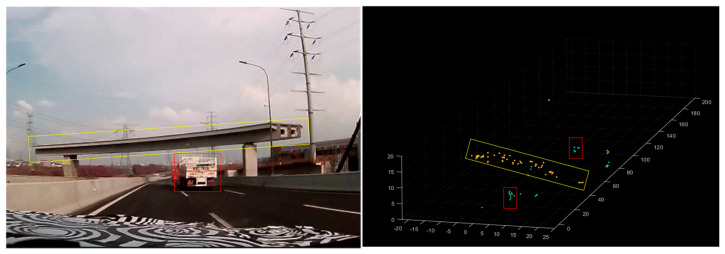
Rendering of the pedestrian bridge in relation to the road vehicles. The red box indicates two moving vehicles, while the yellow box marks the reflection point of the pedestrian bridge. The green dots indicate ground-level targets, while the yellow dots indicate aerial targets at high altitudes.

**Figure 17 sensors-25-04640-f017:**
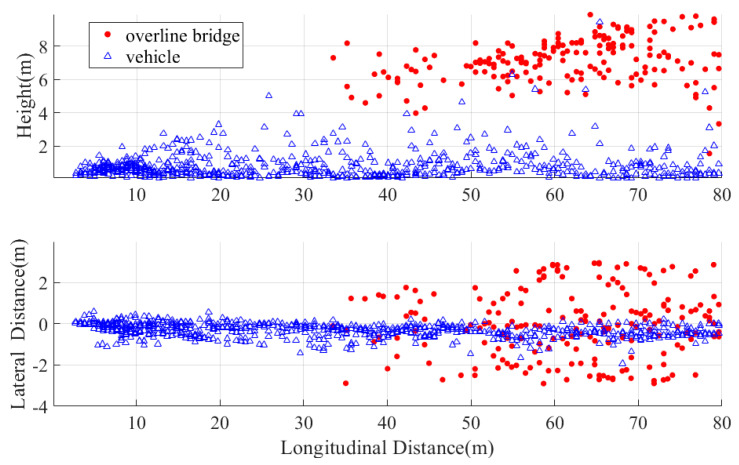
Variation in vehicle/overpass height and lateral distance with longitudinal distance.

**Figure 18 sensors-25-04640-f018:**
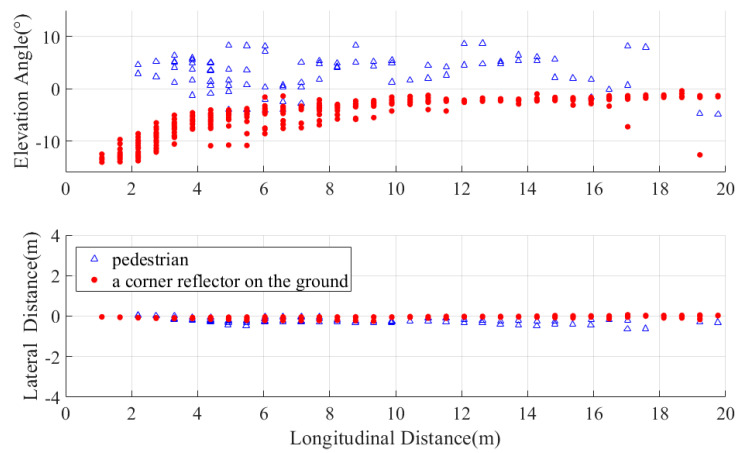
Variation in pedestrian/corner reflector height and lateral distance with longitudinal distance.

**Table 1 sensors-25-04640-t001:** Measured peak amplitude differences in echo signals: normal point clouds vs. noise.

Normal Point Clouds		Noise Point Clouds	
Δp_1-2_	2548	Δp_1-2_	521
Δp_1-3_	2944	Δp_1-3_	736
Δp_1-4_	3560	Δp_1-4_	832

**Table 2 sensors-25-04640-t002:** Statistics of false recognition rates for correct targets using the proposed algorithm.

Target	Number of Tests	Total Points	Marked Points	Recognition Rate (%)	Average Recognition Rate (%)
Pedestrian	first	164	3	1.83	3.65
	second	204	13	6.37	
	third	182	5	2.75	
Car	first	415	29	6.99	5.19
	second	520	22	4.23	
	third	322	14	4.35	
Two-wheel cart	first	322	10	3.11	3.66
	second	432	21	4.86	
	third	332	10	3.01	

**Table 3 sensors-25-04640-t003:** Statistics of correct recognition rate for point clouds caused by noise.

Target	Number of Tests	Total Points	Marked Points	Recognition Rate (%)	Average Recognition Rate (%)
Multipath points caused by cars	first	62	56	90.32	87.60
	second	56	48	85.71	
	third	68	59	86.76	
Ground clutter	first	288	201	69.79	73.41
	second	304	230	75.66	
	third	321	240	74.77	

## Data Availability

Data will be made available on request.

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
