# Peer review of "Development of Low-Cost Single-Chip Automotive 4D Millimeter-Wave Radar"

_sensors, 2025, doi:10.3390/s25154640_

Round 1

Reviewer 1 Report

Comments and Suggestions for Authors

The manuscript presents a comprehensive development and validation of a single-chip 4D millimeter-wave radar system for automotive applications. It addresses important aspects such as cost reduction, clutter suppression, signal processing algorithms, and practical performance in real-world scenarios. It would be beneficial for the paper to be revised as stated in the following comments:

- The quality of presentation is really poor. For example, figure 1 has a large image of polar cordination, but it goes with very small legends and ticks. Similar problem to the legends in Figure 3. Please revise carefully the figures to ensure better quality. Furthermore, several figures and tables are not cited in the correct numerical order (e.g., Figure 4, Table 1, Table 2).

- One additional problem happens to the equations. The equations you provided are not numbered, which makes them hard to follow. Please number them correctly.

- The manuscript consists of some comments which I think accidentally added by someone who revised it before submitting. Make sure you format your paper with the correct MDPI template.

- Your description of the methods needs to be much clearer. Specifically, in Section 3.1, the mathematical derivation for angle estimation is very difficult to follow. I suggest reconsidering this section and adding some more details to define the variables in your equations and provide a more intuitive explanation of how the elevation array is formed and used for measurements. 

- In Section 2, you claim specific performance improvements for your antenna design (e.g., 28% increase in radiation area). These are interesting results, but you need to show the evidence. Please include the comparative simulation results that support these numerical claims.

Author Response

C1:The quality of presentation is really poor. For example, figure 1 has a large image of polar cordination, but it goes with very small legends and ticks. Similar problem to the legends in Figure 3. Please revise carefully the figures to ensure better quality. Furthermore, several figures and tables are not cited in the correct numerical order (e.g., Figure 4, Table 1, Table 2).

R1:

Thank you for pointing this out. We have ‌implemented the following enhancements‌ to address your concerns:

  • ‌Optimized visualization‌ of all  figures with poor quality 
  • ‌Added citations‌ for all previously unciteded figures and tables
  • ‌Conducted a systematic review‌ to ensure ‌all figures and tables are clearly presented and properly referenced throughout the manuscript.

C2:One additional problem happens to the equations. The equations you provided are not numbered, which makes them hard to follow. Please number them correctly.

R1:Thank you for pointing this out. We have ensured ‌proper numbering of all formulas throughout the manuscript‌, executed ‌in strict compliance with the journal's formatting guidelines‌.

C3: The manuscript consists of some comments which I think accidentally added by someone who revised it before submitting. Make sure you format your paper with the correct MDPI template.

R3:Thank you for pointing this out. We have ‌removed all comments‌ from the manuscript to ‌ensure an unobstructed reading experience.

C4:Your description of the methods needs to be much clearer. Specifically, in Section 3.1, the mathematical derivation for angle estimation is very difficult to follow. I suggest reconsidering this section and adding some more details to define the variables in your equations and provide a more intuitive explanation of how the elevation array is formed and used for measurements. 

R4:

Thank you, we agree with your comments and have revised Section 3.1 as follows:
‌All obscure formula derivations have been removed‌, replaced with ‌newly optimized equations‌ that enhance clarity.
‌Complementary enhancements include‌:

  • ‌Explicit term definitions‌ for all first-occurrence variables
  • ‌Expanded textual explanations‌ contextualizing each equation

To implement your recommendations, we have enhanced ‌Section 3.1‌ with detailed explanations, including:‌

  • Specific array layouts;
  • Construction methods‌ for both horizontal and elevation arrays;
  • Corresponding angle measurement procedures.

C5:In Section 2, you claim specific performance improvements for your antenna design (e.g., 28% increase in radiation area). These are interesting results, but you need to show the evidence. Please include the comparative simulation results that support these numerical claims.

R5:Thank you for pointing this out. We acknowledge that relevant simulation results are currently unavailable, as the antenna in question is a commercially produced end-product. Performance data was sourced exclusively from antenna technical documentations. To ensure academic rigor and prevent insufficient evidentiary support in the paper, we propose omitting the comparative data. Kindly confirm whether this modification is acceptable.

Reviewer 2 Report

Comments and Suggestions for Authors

This paper proposes a low-cost single-chip 4D millimeter-wave radar design scheme to address the issue of the lack of height information in traditional 3D radars. My comments are as follows,

1. How can a sparse array with only 4 transmitting and 4 receiving channels balance angular resolution (3°@100 meters) and sidelobe suppression? Can non-uniform array layout further optimize this?
2. The raindrop velocity threshold of ±1m/s is based on the assumption of a maximum inclination of 30°. Under strong wind conditions, the horizontal velocity may exceed the threshold. How can we avoid mistakenly filtering out real low-speed targets?
3. Will the noise suppression algorithm in a dense multi-target scenario  cause missed detections due to peak interference?
4. The pitch angle error of 0.4°@±13° corresponds to a height error of approximately ±0.9 meters at a distance of 100 meters. How can we ensure precise detection of low obstacles ?
5. Can the 360MHz DSP + HWA accelerator handle real-time processing of multi-target tracking (>100 point clouds/frame) and complex algorithms (raindrop statistics + noise filtering)?
6. The suppression scheme for interference between radars operating on the same frequency has not been discussed. How can stability be guaranteed in multi-vehicle cooperative scenarios?
7. In Figs. 12, 13, 14, and 15, if the test results need to be presented, please do so in a clearer way, such as using curves, to allow readers to see the content. If it is not necessary to show them, they can be deleted.

Author Response

C1:How can a sparse array with only 4 transmitting and 4 receiving channels balance angular resolution (3°@100 meters) and sidelobe suppression? Can non-uniform array layout further optimize this?

R1:

Thank you for your comments. Regarding this issue, we have enhanced ‌Section 3.1‌ with detailed explanations, including:‌ 1)Specific array layouts;2)‌Construction methods‌ for both horizontal and elevation arrays;3)‌Corresponding angle measurement procedures;4)‌Array resolution metrics‌ with ‌sidelobe characteristics.

We balanced ‌horizontal/elevation performance‌ and ‌sidelobe control‌ under non-uniform array configurations through deliberate trade-offs. After multiple optimizations, the current version represents our ‌best iteration‌ with ‌minimal room for further improvement‌.

The intentionally elevated sidelobe levels stem from our design choice to ‌prioritize resolution and angular accuracy‌. Corresponding mitigation measures in subsequent processing effectively suppress sidelobe impacts, yielding: 1)Controlled clutter levels‌ in practical operation; 2)Verified effectiveness‌ in field applications

C2:The raindrop velocity threshold of ±1m/s is based on the assumption of a maximum inclination of 30°. Under strong wind conditions, the horizontal velocity may exceed the threshold. How can we avoid mistakenly filtering out real low-speed targets?

R2:Thank you for pointing this out. In our practical implementation, ‌the strategy additionally incorporates multi-parameter filtering‌ – including power, SNR, noise floor, RCS, etc. – coupled with ‌scenario-specific processing protocols‌. This integrated approach robustly preserves target authenticity while eliminating clutter interference.

C3:Will the noise suppression algorithm in a dense multi-target scenario  cause missed detections due to peak interference?

R3:Thank you for pointing this out. Our strategy formulation ‌presupposes‌ that: 1)Clutter detection probability‌ must reach ≈50%;2)False alarm rate‌ for actual targets remains ≤10%. The judgement thresholds were ‌empirically derived‌ from extensive field data analysis to meet these benchmarks. With typicallysufficient target sampling points‌, field tests demonstrate ‌minimal operational impact‌ in actual road tests.

C4:The pitch angle error of 0.4°@±13° corresponds to a height error of approximately ±0.9 meters at a distance of 100 meters. How can we ensure precise detection of low obstacles ?

R4:Thank you for your comments. The ‌0.4° maximum angular error‌ applies specifically at the ‌beam edge‌, whereas ‌near 0° azimuth targets achieve significantly higher precision‌. Additionally, we have addressed on ‌Page 16‌ that under ‌ground multipath effects‌ and ‌limited array resolution‌, systems may misclassify ‌high-altitude targets beyond 80m as ground objects‌. This limitation can only be mitigated with ‌additional channels‌.

C5:Can the 360MHz DSP + HWA accelerator handle real-time processing of multi-target tracking (>100 point clouds/frame) and complex algorithms (raindrop statistics + noise filtering)?

R5:Thank you for your comments. The processor integrates a ‌360MHz DSP + HWA+ ‌300MHz MCU‌(primarily tasked with executing tracking algorithms), robustly sustaining ‌real-time processing of: 1)≥32 tracked targets; 2)≥300 point clouds per frame – including all algorithmic workloads – within a ‌70-80ms frame cycle‌.

C6: The suppression scheme for interference between radars operating on the same frequency has not been discussed. How can stability be guaranteed in multi-vehicle cooperative scenarios?

R6:Thank you for raising this point. ‌Regarding anti-interference measures, we currently have no dedicated solutions. It should be noted that our system employs standard countermeasures‌ – specifically ‌frequency hopping‌. Consequently, ‌this aspect was deliberately omitted from the current scope of the paper.

C7:In Figs. 12, 13, 14, and 15, if the test results need to be presented, please do so in a clearer way, such as using curves, to allow readers to see the content. If it is not necessary to show them, they can be deleted.

R7:Thank you for pointing this out. Following your suggestion, we have incorporated quantitative results to our presentation(see pp. 16-17). This allows readers to see ‌not only‌ the intuitive and explicit outcomes ‌but also‌ the quantified metrics, providing a more clearer understanding of our solution.

Round 2

Reviewer 1 Report

Comments and Suggestions for Authors

The response is fine.

Author Response

Thanks again for reviewing this!

Reviewer 2 Report

Comments and Suggestions for Authors
  1. The review mode of the manuscript should be turned off.
  2. The annotations and comments in the manuscript should be deleted.
  3. All the formulas in the text need to be numbered.

Author Response

C1:The review mode of the manuscript should be turned off.

R1:Thank you for  your comments. We have turned off the review mode in the latest revised manuscript.

C2:The annotations and comments in the manuscript should be deleted.

R2:Thank you for  your comments. We have deleted the annotations and comments in the latest revised manuscript.

C3:   All the formulas in the text need to be numbered.

R3:   Thank you for  your comments. We have checked the entire manuscript to ensure all formulas are numbered in the latest revised manuscript.